# Shifting Waters: The Challenges of Transitioning from Freshwater to Treated Wastewater Irrigation in the Northern Jordan Valley

Mohamed Hassan Tawfik [1,2,*], Hadeel Al-Zawaidah [1], Jaime Hoogesteger [2], Maha Al-Zu'bi [1], Petra Hellegers [1], Javier Mateo-Sagasta [3] and Amgad Elmahdi [4]

[1] International Water Management Institute, MENA Region, Nile St, Giza 12111, Egypt
[2] Water Resources Management Group, Wageningen University, 6708 PB Wageningen, The Netherlands; jaime.hoogesteger@wur.nl (J.H.)
[3] International Water Management Institute, Headquarters, 127, Sunil Mawatha, Battaramulla, Colombo 10120, Sri Lanka
[4] Green Climate Fund, Songdo, Incheon City 22004, Republic of Korea
* Correspondence: mohamedhassantawfikhassan.anis@wur.nl

**Abstract:** Jordan's water scarcity prompted a national plan whereby treated wastewater is utilized to amend agricultural irrigation water so as to reallocate freshwater to urban/domestic uses. The policy, however, has engendered farmers' resistance in the Northern Jordan Valley (NJV), causing a stalemate in putting new infrastructure into operation. This research investigated the socio-economic causes of farmer resistance and contestation, and examined the government's institutional approach to overcome the challenges. We found that the perceived risks of wastewater reuse such as salinization and restrictions from international markets figure prominently in the farmers resistance. As yet, farmers have managed to avoid the shift to treated wastewater use by using the political agency of elite farmers who control the Water Users Associations. These same farmers have adopted informal water access practices to overcome freshwater shortages. At the same time, small producers who don't have possibilities to access extra water and with less political clout seem more willing to irrigate with treated wastewater. We conclude that understanding the heterogeneous context in which the envisioned wastewater users operate is key to predicting and solving conflicts that arise in treated wastewater reuse projects.

**Keywords:** Northern Jordan Valley; wastewater; reuse; water reallocation; water user association; water policy; Middle East; Jordan

## 1. Introduction

The Middle East and North Africa (MENA) region is facing a growing gap between water supply and demand [1]. Urban water demand is increasing, with cities often being granted priority in freshwater allocations, at the expense of irrigated agriculture. Given that agriculture is the MENA region's largest water consumer, although it also has the lowest economic return on water use, this sector faces a looming challenge in sustainably meeting its current and future water needs [2,3]. Governments in the MENA region are attempting to amend the gap in irrigation water availability by promoting the reuse of treated wastewater [4,5].

Over the past decades, Jordan has experienced a drastic reduction of surface water availability, growing water demand, especially in agriculture and for urban uses, and over-abstraction of groundwater resources [6,7]. By 2025, the country's overall water shortage is expected to reach 1.521 million cubic metres (MCM) per year [8]. This scarcity is aggravated by geopolitical challenges, which have stood in the way of a fair distribution arrangement for transboundary water [6,7,9].

Against this background, Jordan is seeking to maximize its treated wastewater reuse capacity to reach 80% reuse of the generated wastewater by the year 2050, up from 30% at present [8]. Treated wastewater reuse in agriculture would also provide a nutrient-rich source of water for fertigation that could help reduce the negative impacts of inorganic fertilizers on the environment [10–12]. Reaching this target, however, will not be easy, as it requires huge investments in infrastructure, a willingness of actors in the agricultural sector to adopt water reuse practices and the ability to cope with the high running costs of wastewater treatment plants, especially as a result of energy consumption [13].

In practice, substantial steps are still needed to achieve the shift towards greater water reuse capacity. However, Jordan's institutional landscape has been evolving towards this goal since the 1970s, when the government issued its first treated wastewater quality standards under Law 21 [14]. In the early 2000s, Jordan's water reuse policy was placed under its 'Water for Life' strategy, in which a key focus is assuring the quality of treated wastewater and its safe use in terms of public health and environmental protection [15]. The strategy targets the expanded use of treated wastewater in agriculture to help meet the water demands of major irrigated agriculture schemes. Under the more recent 'Water Green Growth Action Plan 2021–2025' [16] water reuse expansion is a key target.

With two recent policies, the Water Substitution and Reuse Policy and the Water Reallocation Policy [17], Jordan's government aims to reallocate part of the agricultural sector's share of freshwater towards the municipal sector. This will provide drinking water for city residents while at the same time providing nutrient-rich treated wastewater from cities to amend the water deficit in agriculture. However, farmers in the Northern Jordan Valley (NJV)—one of the country's foremost agricultural areas—view the plan as a threat to their traditional agricultural practices and livelihoods. Farmers' resistance has continued to stall the further implementation of the water reuse–reallocation scheme. Though the needed hydraulic infrastructure has been in place since 2017, the scheme has not been operationalized. This delay jeopardizes the plan's potential to contribute towards solving Jordan's severe water and agricultural sector challenges.

This paper builds on the works of Mollinga (1998) [18] and Uphoff (1986) [19] by advancing a holistic understanding of the coexistence of formal and informal practices as integrally intertwined components that shape and are shaped by the socio-economic context governing irrigation water allocation and management in the Jordan Valley. Adopting this theoretical lens, our aim is to understand the role of informal practices in facilitating farmers' access to water for irrigation while they resist the government's plans. At the same time, we seek to understand how stakeholders perceive the plan to reuse treated wastewater in agriculture and reallocate freshwater resources.

First, this paper presents the qualitative research methodology employed to collect and analyse the available data, alongside the theoretical orientation of the analysis and the study area description. It then discusses the interview results with farmers and formal stakeholders regarding irrigation water availability, access to water and the in/formal institutional arrangements by which irrigation water is allocated and current practices to access irrigation. Farmers were also asked about their perceptions regarding the shift to the use of treated wastewater for irrigation, and whether they would accept this shift or not and why. The discussion section zooms in on how farmers have managed to resist the reuse–reallocation plan. The paper concludes with a recommendation for an inclusive and participatory approach to bridge the trust gap between farmers and government and help build an inclusive water reuse plan with minimized risk to farmers in the NJV.

## 2. Materials and Methods

This research is based on qualitative methods involving document review and interviews with key informants and farmers. The reviewed documents included academic literature and official reports addressing water resources management challenges in Jordan, from both a technical and an economic perspective. Reports from donor organizations and government water policy documents were key to understand the government's plans for

water resources management and reuse expansion. The informant and farmer interviews were semi-structured and involved representatives from Jordan's water and agricultural sector (including professors, researchers and government officials), farmers from the NJV, and members of one water user association (WUA) in the NJV.

Stakeholder identification and interviews were conducted under the umbrella of the ReWater MENA project, which took place in the NJV and was implemented by the Royal Scientific Society (RSS) in coordination with the International Water Management Institute (IWMI). Selection of farmers for interviews was facilitated by a local community-based organization (CBO) that was active around farm units adjacent to irrigation lines 11, 12 and 13. The selection process was random, based on farmers' availability and their willingness to participate in the interviews. This resulted in interviews with 18 NJV farmers, including WUA board members of irrigation lines 12 and 13.

The key informant (expert) interviews were conducted in Amman with representatives of relevant governmental and non-governmental institutions, such as the Jordan Valley Authority (JVA) (we interviewed the director of the division in charge of WUA organization), the RSS and the IWMI office in Amman. The semi-structured interviews were conducted over three months (June, July and August 2021). Their focus was to understand stakeholders' views on irrigation water challenges in the NJV, the on-farm practices used by farmers to manage irrigation water quotas and overcome scarcity, the presumed role of water reuse in the agricultural future of the NJV, and the role of WUAs in irrigation water allocation in the NJV, as well as whether or not WUAs would have a role in facilitating treated wastewater reuse expansion and acceptance in the NJV.

### 2.1. Theorizing Water Reuse and Reallocation in the NJV

Irrigation water transfer and allocation among users (henceforth irrigation water management) often takes place through top-down formal arrangements in which various technological, infrastructural, institutional and organizational interventions are set to control water flow from source to user according to codified rules and agreements [15]. Parallel to and intersecting with those formal arrangements, there are co-existing informal arrangements that local communities adopt as adaptive mechanisms, to adjust to the impacts of formal arrangements for access to irrigation water [20]. Informal arrangements are often labelled as illegal practices [21], although this is not always the case. Informal arrangements are socially embedded practices that have been forged by the local context (socio-economic and everyday political factors), and frequently they are shaped by the formal arrangements themselves [18].

The interaction between formal and informal arrangements reflects the complexity of the irrigation water management process. That process encompasses government strategic plans, infrastructural projects, donor interventions, organizational and institutional reforms, and farmers' acceptance of or resistance to all these, as well as farmers' interest in maximizing their profitability by accessing more irrigation water through various informal practices which many governments fail to codify or regulate.

To grasp this complexity and dynamics, this paper combines Mollinga's (1998) and Uphoff's (1986) frameworks in order to understand the interaction of formal and informal irrigation water management practices in the NJV. Both frameworks have in common the concept of *water control* and the understanding of irrigation as an *inherently political and thus highly contested practice*. In a context of scarcity, as in Jordan, irrigation water becomes a contested resource between various stakeholders, with these stakeholders typically having asymmetric power and influence on access to irrigation water. In such cases, each stakeholder group aims to maximize their profits, either through formal arrangements (e.g., new rules and regulations or construction of water control infrastructure) or by adopting informal arrangements to resist and circumvent those formal interventions. Uphoff's (1986) framework would have been sufficient for this paper if the aim was to understand the dynamics of water control activities and the role of farmers in managing those activities regardless of the agricultural arrangements in the study area. Combining Uphoff's frame-

work with Mollinga's enables us to position water control activities in the NJV within the existing agro-political ecology context of the Jordan Valley (Figure 1). Indeed, this context shapes perceptions of irrigation water quality, quantity, reuse and reallocation mechanisms among the different stakeholders and users within and outside the NJV.

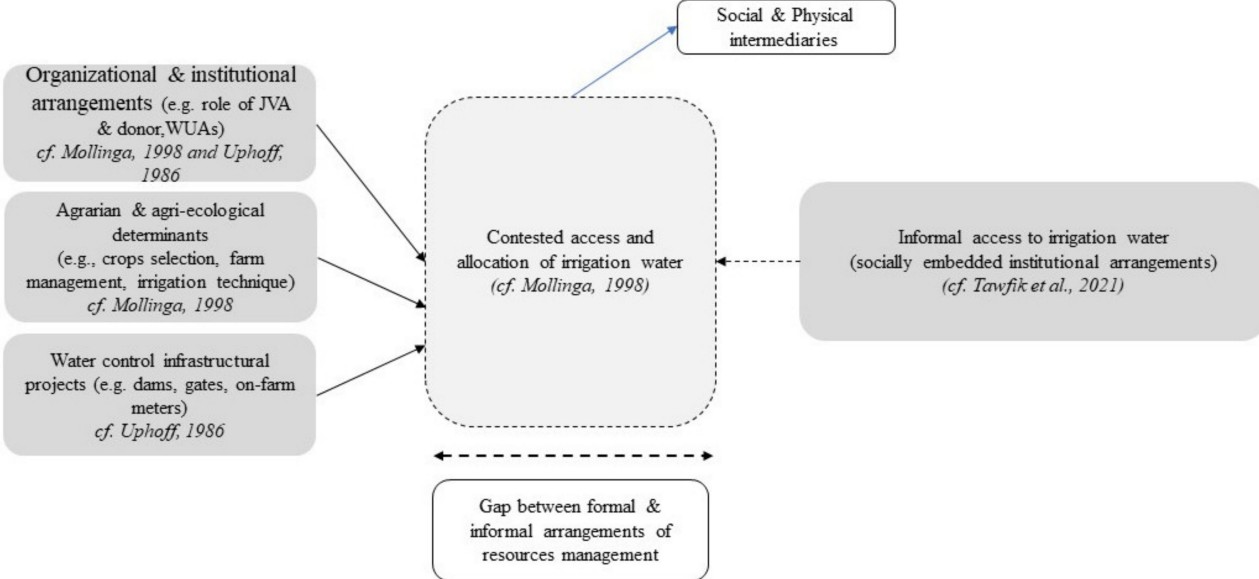

**Figure 1.** Combining Mollinga's (1998) and Uphoff's (1986) analytical frameworks [18,19,22].

In the framework (see Figure 1), the left-hand side represents interventions and activities that take place through formal arrangements initiated either by donor organizations or governments. Those activities reflect the material and social conditions of possibilities to enable irrigation water management in the NJV, as well as the water flow control that takes place through organizational/institutional means and infrastructural or technological means.

The right-hand side of the framework represents the various informal (socially embedded or socio-normative) practices that farmers in the NJV adopt to circumvent and resist the formal arrangements [22]. Similar to the formal arrangements, informal practices can target either the physical or the social conditions of possibilities for irrigation water management.

Finally, at the centre of the framework there is the social space where the formal and informal arrangements interact over the contested irrigation water resource in terms of water quantity, quality and water quota allocation timing (as in the case of the NJV). According to Mollinga (1998), this interaction is mediated by various social and physical factors at different scales, represented by the dashed border of the central block.

### 2.2. Study Area

The Jordan Valley is one of Jordan's three agro-ecological zones. The three zones (the Jordan Valley, desert and highlands) are heterogenous in terms of biophysical characteristics (sources and quality of water, micro-climate, soil types and crops), as well as the local social, economic and political context that shapes agricultural activity in each area [23] (Table 1). The Jordan Valley is the largest among the three zones in terms of surface area and irrigation water consumption (it consumes around 25% of the available water resources in Jordan), and it is considered the breadbasket of the country, thanks to its close proximity to water sources, its altitude (200–400 m below sea level) and its soil fertility [24].

**Table 1.** Agricultural activities in the Jordan Valley [23].

| Agro-Ecological Zone | Crops | Source of Irrigation Water |
|---|---|---|
| Northern Jordan Valley (NJV) | Citrus<br>Banana<br>Vegetables | Surface water: from King Abdullah Canal (KAC)<br>Rainfed |
| Middle Jordan Valley (MJV) | Vegetables (greenhouses) | Mixed water: treated wastewater + freshwater from KAC |
| Southern Jordan Valley (SJV) | Vegetables<br>Date palm trees | Groundwater wells |

The NJV is divided into agricultural basins that receive weekly irrigation water quotas through a pressurized or gravity pumping network. The Jordan Valley Authority (JVA) oversees irrigation water flow management and allocation processes. Each basin consists of a group of agricultural units sharing the same irrigation pump and water distribution network. The units vary in size from 30 to 40 donums (3–4 ha), with farmers owning or renting one or more agricultural units in the same or in different basins.

Despite its prominence as one of the country's foremost irrigated agriculture areas, agriculture in the Jordan Valley is threatened by severe physical water scarcity as a result of climate change impacts, transboundary water allocation challenges and increasing demand for water resources in other sectors [6,25]. Concurrently, the agricultural importance of the Jordan Valley is affected by the government's perception of the agricultural sector here as being a 'wasteful' consumer of water resources, as it usurps 50–70% of the country's freshwater withdrawals [7]. Furthermore, primary agriculture's contribution to GDP is falling (i.e., excluding the full agricultural value chain that extends to activities in the broader agrifood sector), having dropped from 8.1% in 1991 to 3.6% between 2011 and 2016 [26,27].

Treated wastewater from the As Samra wastewater treatment plant has been used to partially substitute freshwater for farmers in the MJV (complying with the Jordanian reuse standards). Here, treated wastewater is collected and mixed with freshwater at the King Talal Dam reservoir, followed by another mixing stage with freshwater from the Yarmouk River, Peace Conveyor and Mukheiba wells, before water reaches farmers to meet their irrigation needs. This has facilitated the reallocation of freshwater to help meet increasing urban water demand through water reallocation infrastructure linking the rural MJV and urban settings (e.g., Amman). This has served to counterbalance rising water demand in the nation's capital and major cities. Some 50 million cubic meters (MCM) per year of freshwater were transferred from the Jordan Valley to Amman in the early 2000s, and plans are in place to increase this amount to 90 MCM per year by 2025 [24].

Treated wastewater reuse and freshwater reallocation plans for the NJV date back to 2001, but these have not been operationalized—despite the area being identified as a top priority for water reuse and freshwater reallocation [28]. Instead, irrigation in the NJV has continued to rely on freshwater from the King Abdullah Canal (KAC). Farmers in the NJV have a long tradition of citrus plantations, with the fruits destined for export to regional markets in Syria and the Gulf States, analogous to the earlier practices of farmers in the MJV before the shift to reused water for irrigation. Farmers in MJV have since lost their citrus trees, attributing that loss to the poor quality of the treated wastewater from the As Samra plant, particularly before the plant upgraded to mechanical treatment in 2007 [29,30]. The shift to treated wastewater reuse and the corresponding change in water quality thus led to a shift in crop selection across the MJV to crops that are more resilient and less sensitive to salinity [31]. Despite significant improvements in the quality of the treated wastewater from the As Samra plant, NJV farmers have continued to reference the MJV case as justification for their fear and resistance to receiving mixed treated wastewater for irrigation.

## 3. Results

This results section explores water allocation and access in the NJV in a structure corresponding with the aspects of irrigation water management presented in the theoretical framework (see Figure 1).

### 3.1. Irrigation Water Organizational and Institutional Arrangements in the NJV

The Jordan Valley Authority (JVA) is responsible for water quota allocation from the KAC to farmers across the Jordan Valley, either directly through the JVA's canal operators or through water user associations (WUAs). In 2001, farmers' involvement in irrigation water management was facilitated with the support of GIZ. The first WUA involving farmers was established in 2002. It was called "the Jordanian corporative" and chaired by Jordan's minister of agriculture. The idea of farmer involvement was further developed based on examples from different countries where visits took place over time. The first was to Damanhur, Egypt, in 2004, followed in 2005 by visits to Syria and Turkey.

Until 2007 work in the corporative was voluntary and unpaid. In 2008, two "entrepreneur" locations were selected in the NJV, near pumping station 33. Here, a transfer of responsibilities agreement was signed, including both water distribution and censorship responsibilities (monitoring for violations and reporting these to the JVA). In an interview, the director of the JVA division in charge of WUA organization stated that the "observed positive impact of WUAs on irrigation water management" had encouraged JVA to expand the number of such agreements, which reached five in 2009. Later, in 2015, a new transfer of responsibilities agreement was signed in which WUA employees would receive payment from the JVA. They were also made responsible for regular maintenance, although emergency maintenance and repairs remained the responsibility of the JVA. The agreement also included definitions of technical and geographic regions in which the contract was valid and the precise responsibilities of both the JVA and the signatory associations. WUA responsibilities included water distribution, pumping station operation and irrigation water network maintenance.

In 2020, 18 WUAs had active agreements with the JVA. In that year, WUAs covered 18% of the area served by irrigation in the Jordan Valley as a whole. Despite an increasing number of WUAs, they had not developed into fully financially and/or legally autonomous entities as initially foreseen [32]. The challenges the WUAs faced were reflected in the responses of the interviewed WUA members, as they noted the associations' limited capacity to take a leading role in irrigation water management due to their financial and legal dependency on the JVA. Fourteen of the farmers interviewed also commented on the WUAs' lack of capacity for conflict resolution and their inability to ensure that farmers received their allocated water quotas. They stated that the WUAs had had little positive impact on irrigation water management in the NJV, and they mainly benefited rich farmers and WUA board members (the majority of whom were rich farmers). In this context, they used the word "rich" to refer to farmers with access to financial means (e.g., through inheritance or side jobs other than farming) and with access to knowledge regarding new farming technologies (e.g., water storage in fishponds, drip irrigation, the use of solar-powered water pumps).

The interviewee from the JVA division agreed that the WUAs faced challenges and there was room for improvement. Improvements were said, however, to be taking place. For example, in North Shuna (in the NJV) six associations had been incorporated into one association called "Water Users Association in North Shuna". It was established as a voluntary association (non-profit) under the Ministry of Social Development, which reduced the cost of the WUA, as the establishment of a commercial association requires a payment of 360 Jordanian dinars (JOD) (USD 514). The Ministry of Water and Irrigation manages the association through a service agreement instead of an employment agreement.

In addition to the JVA and WUAs, donor agencies (particularly the United States Agency for International Development (USAID)) have had an influential role in irrigation water management and water allocation in the NJV. In 2001, the USAID's Water Resource

Policy Support Project identified the NJV as one of three priority areas in the Amman–Zarqa Basin for implementation of a reuse–reallocation plan [28]. A USAID report states that the NJV provides "very attractive" potential to divert large amounts of freshwater to Amman at a relatively low cost. Some 57 MCM was estimated as the amount that could potentially be transferred annually, slightly exceeding Jordan's freshwater allocation under to the 1994 peace treaty with Israel [33].

Despite the organizational and technical potential to secure a more sustainable water supply for rural and urban users, the report acknowledges that the reuse–reallocation plan would have "considerable" socio-economic impact on farmers in the NJV. Particularly, the shift to reused water would affect cropping patterns, due to the change in water quality. Farmers therefore faced potential losses or reductions of income, especially if they lost their traditional citrus plantations [28], as occurred in the MJV. The interviewed farmers mentioned this as a key concern.

Other than the KAC, alternative sources of irrigation water in the NJV were limited to shallow groundwater wells (farmers call these 'springs') and deep groundwater wells, though closer to the Jordan River these wells were often saline. All of these alternative sources were regulated by JVA through the issuance of permits for groundwater wells and fines in cases of violations (e.g., illegal wells). Obtaining permission for a groundwater well cost JOD 10,000–12,000 (USD 14,000–28,000), in addition to the cost of digging the well (approx. JOD 20,000 or USD 28,000), making this an unaffordable option for ordinary farmers, most of whom were poor.

*3.2. Agrarian and Agro-Ecological Determinants*

The key agrarian and agro-ecological determinants are crop selection, farm management and on-farm water management, including irrigation technique and the farm water quota according to the regulations in the NJV. Most of the interviewed farmers from the selected basins (12 farmers) owned citrus plantations, and five interviewees were hired operators with long experience in farm management. One interviewee was a 'guarantor', an arrangement similar to a rental agreement in which the farm owner receives a fixed monetary sum and the guarantor takes over farm management and crop marketing.

The interviewed farmers stated that many farm owners in the NJV preferred managing their farms through guarantors, to avoid the increasingly challenging marketing context and other uncertainties that threatened agricultural productivity and farm revenue. Table 2 presents the characteristics of the farms selected for this study in the NJV regarding crop selection, on-farm irrigation water management, water allocation mechanism and alternative sources of water for irrigation through informal access.

Irrigation water quotas in the NJV were crop-based, in which larger quotas were allocated to farmers growing citrus and bananas compared to farms growing vegetables (Table 3). Vegetables were mainly grown during the winter season under rainfed conditions. Citrus and banana farmers received roughly twice the water allocation of vegetables. The JVA managed these allocations. Among its responsibilities, the JVA issued permits to farmers who were willing to grow high-water quota crops (mainly citrus and bananas). Despite stimulus policies being in place to reduce irrigation water consumption, the JVA had legalized several unregistered citrus plantations that had begun operating between 1991 and 2002 [34]. These farms consequentially became legally entitled to receive the higher water quotas for citrus farms.

**Table 2.** Characteristics of the farms in the Northern Jordan Valley (NJV) randomly selected for this study.

| Farm Management | |
| --- | --- |
| -    Farm operation | 12 farm owners, 5 managers/operators, 1 guarantor |
| -    On-farm irrigation water storage | 11 farms had on-farm water storage structures (i.e., earthen or concrete ponds)<br>7 farms irrigated directly without storage |
| -    Irrigation technology | 15 farms used irrigation hoses<br>3 farms used drip irrigation |
| Irrigation water allocation mechanism | |
| -    Jordan Valley Authority (JVA) (canal operators) | 14 farms received water quotas through the JVA |
| -    Water user associations (WUAs) | 2 farms received water quotas through WUAs |
| -    Other | 2 farms did not have access to a formal water quota (their land plots were not officially registered as agricultural lands) |
| Alternative source of water for irrigation (informal access) | 11 farms reported having no access to water sources other than their water quotas<br>4 farms reported using alternative or additional water sources either sporadically or continuously, such as shallow or deep wells, water abstracted from the Jordan River and illegal abstraction from the KAC<br>3 farms requested extra water quotas (extra hours of irrigation) from the JVA when water shortages arose |
| Crop types | 12 citrus farms<br>3 vegetable + citrus farms<br>1 grape farm<br>1 citrus + grape farm<br>1 citrus + date palm tree farm |
| Average agricultural units per interviewed farmer | 1–2 units per interviewed farmer, with average unit size in the NJV being 33 donums (3.3 ha) |

**Table 3.** Crop-based water quota allocations in the Northern Jordan Valley (NJV), citrus versus vegetable farms [35].

| Period | Citrus Farm Water Allocation (Average m$^3$/ha/day) | Vegetable Farm Water Allocation (Average m$^3$/ha/day) |
| --- | --- | --- |
| High water demand (April to October) | 30 | 15 |
| Low water demand | 20 (extra water allocated upon request) | 15 |

Increasing water scarcity and diminishing water quotas in the NJV had affected farmers' selection of irrigation techniques, corresponding to variations in farmers' knowledge and their financial means to address the problem. We observed that rich farmers were capable of shifting to advanced irrigation techniques. Unlike their poorer counterparts, they could gain access to knowledge and financial means to reduce the impacts of water scarcity. Among the interviewed farmers, only three (two from rich families, and one who was a retired JVA employee) had shifted from surface irrigation to drip irrigation with filtration units to avoid system clogging. These three farmers (all growing citrus) reported

significant water savings after installation of the drip irrigation. The remaining 15 farmers were using traditional water distribution hoses fixed at the base of each tree.

During the interviews, half of the interviewees stated that the water scarcity problem in the NJV was the result of mismanagement of the country's water resources, which they said affected farmers across the entire Jordan Valley, though they also noted mismanagement of water quotas by NJV farmers and NJV farmers continued use of traditional irrigation techniques. Half of the interviewees stated that Jordan was indeed suffering from a reduction in physical water availability.

Farmers' perceptions of the potential of treated wastewater reuse for irrigated agriculture were largely influenced by changes in the MJV. Half of the interviewed farmers based their perceptions of reused water for irrigation on the experiences of farmers in the MJV. As noted, the latter attributed loss of their citrus plantations to the low quality of treated wastewater from As Samra (particularly, high salinity). Additionally, three farmers reported they had access to treated wastewater during a trial conducted by the JVA in 2014, in which treated wastewater was mixed with freshwater from the KAC. Those farmers said the water had smelled bad and some field workers had suffered a skin rash. The trial continued for two weeks before it was stopped due to farmers' complaints. Despite this, 11 of the interviewed farmers expressed interest in discussing the potential of treated wastewater reuse for crops such as vegetables and date palm, but not for citrus, as these trees were sensitive to salinity.

### 3.3. Water Control Infrastructure

3.3.1. Jordan Valley Level

The Jordan Valley is part of the Lower Jordan River Basin (LJRB), a semi-closed basin in which water demand exceeds the current supply [24]. Intensive irrigated agriculture in the Jordan Valley has been facilitated by availability of water from the King Abdullah Canal (KAC), which was constructed in the 1960s. The KAC receives its water from the Yarmouk River, the Peace Conveyor (in accordance with the 1994 peace treaty with Israel) and groundwater wells.

The NJV receives freshwater for irrigation from the KAC, which extends to the first few kilometres of the MJV. Freshwater from the KAC is then conveyed to Amman for domestic and industrial use (including for the tourism industry). After the diversion point, the KAC receives a different water flow coming from the King Talal Dam (KTD) to the east. The source of this water is the As Samra wastewater treatment plant which collects and treats Amman's and Zarqa's wastewater. The treated wastewater flows to the KAC through the valleys and enters the KAC at a junction point in the MJV (Figure 2). Accordingly, the rest of the MJV relies mainly on treated wastewater for irrigation (the ratio of treated wastewater to freshwater is 6:1) (Table 4).

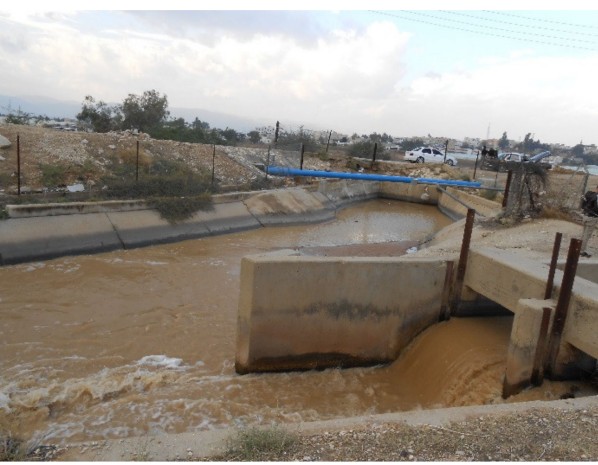

**Figure 2.** Diversion point at MJV.

**Table 4.** Water allocation to the Northern Jordan Valley (NJV) and Middle Jordan Valley (MJV) (source: [23]).

| Water Source | Distributed to | Volume (Mm$^3$/year) | Water Control Infrastructure |
|---|---|---|---|
| King Abdullah Canal (KAC) (from Yarmouk River, Peace Conveyor and groundwater wells | NJV | 55 | KAC + Pump stations |
| Imported from northern wadis | | 15 | Pump stations |
| Treated wastewater from As Samra treatment plant | MJV | 60 | King Talal Dam (KTD) + pump stations |
| KAC | | 10 | KAC + Pump stations |
| Total water allocated to NJV + MJV | 140 Mm$^3$/year | | |
| Total water allocated to Amman from the Jordan Valley | 50 Mm$^3$/year | | |

Additionally, new water control infrastructure was developed in 2017 to reallocate freshwater from the NJV to the city of Irbid for domestic use while supplying farmers in the NJV with treated wastewater for their agricultural activities (in a 6:1 ratio, similar to the MJV) (see Figure 3). This arrangement, however, has not been made operational due to farmers' resistance.

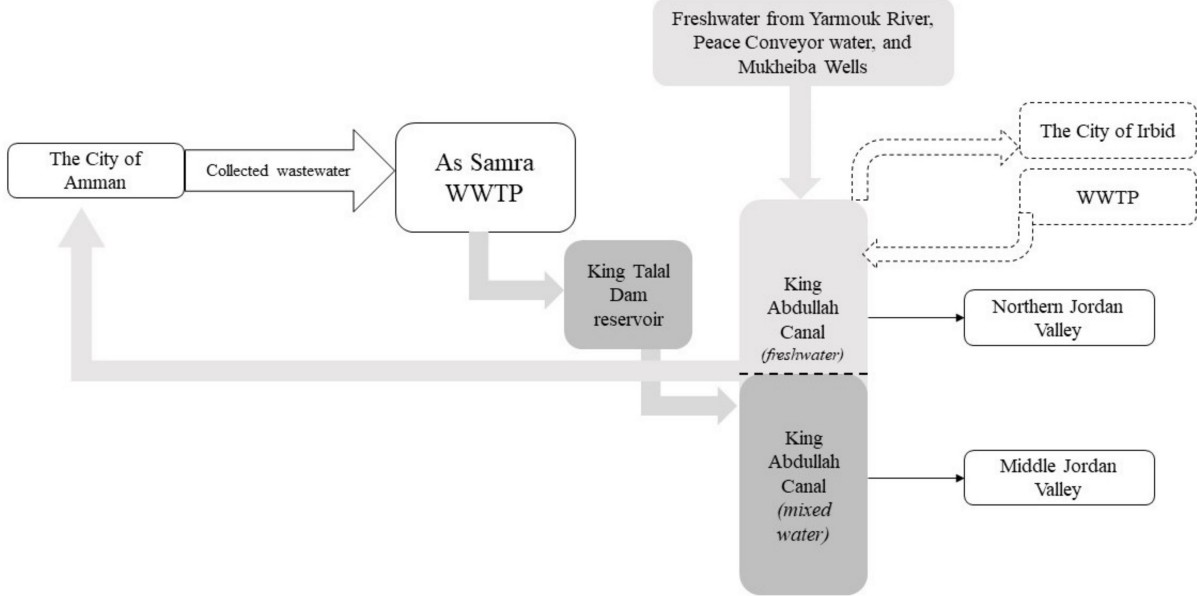

**Figure 3.** Current and proposed water reallocation plans in the Jordan Valley (dashed lines indicate proposed water reallocation scheme). WWTP = wastewater treatment plant.

### 3.3.2. Farm Level

On-farm irrigation water management and distribution in the NJV is regulated by the JVA through installation of a 'Farm Turn-out Assembly' (FTA) at the head of each agricultural unit (i.e., one FTA releases water for 30–40 donums). The JVA equips the FTAs with flow meters to regulate and limit water flows to a rate of 6–9 l per second for an average of 12 h per week for each farm, while also measuring water consumption. The interviewed farmers reported annual irrigation water bills ranging from JOD 80 to JOD 200, in which 'annual' refers to the summer months when farmers rely on pumped irrigation water in the absence of rainfall. Variation in the water bills depended on the number of units each farmer owned. Hence, water bills did not reflect the volumetric

water consumption per agricultural unit, but rather, the number of irrigation units each farmer had.

*3.4. Informal Access to Water for Irrigation*

Freshwater availability and accessibility through informal practices and arrangements was a key element that sustained agricultural productivity in the NJV, and this augmented farmers' resistance to the plan to introduce water reuse for irrigation. According to interviewees, farmers' social network and influence affected their ability to access irrigation water through formal and informal practices. Those farmers who were unable to access additional freshwater, through formal or informal means, were mainly poor farmers with weak connections (e.g., with either the canal operators or WUA members). For them, water shortages had caused a significant diminishment of crop productivity, as reported during the interviews. To adapt to the reduced water quotas, some of these farmers had cut back on the number of citrus trees per farm unit. Farmers with more than one farm unit often chose to combine and reallocate all of their water quotas to a single unit, to meet the irrigation water demand there. At the same time, the rich farmers interviewed reported little or no complaint regarding access to water quotas. They blamed 'other' farmers for mismanaging their water quotas and faulted their continuing use of low-efficiency traditional irrigation techniques. Some of the rich farmers did report at times requesting an extra irrigation water quota from the JVA, which they said would be 'promptly' provided. However, less influential and poor farmers did not report this practice (i.e., requesting and receiving extra water quotas from the JVA). Despite the ready provision of the extra quotas, even these were said to be shrinking due to the diminishing availability of water in the Jordan Valley. Rich farmers tended to fill gaps by abstracting water from deep wells, though these were often saline and required the use of desalination units. Many refrained from applying for formal permission for these wells or registering them, likely considering their social and political influence sufficient to protect them from the 'strict' government fines on unregistered wells [36].

One of the most common practices reported was tampering with FTA units, leading to inequity in water quota distributions between farmers at the beginning (upstream) of the irrigation line and downstream. Such practices had drastically affected the gravity operated irrigation lines (i.e., unpressurized lines not equipped with pumping stations), as farms at the beginning of these irrigation lines or at a lower slope could receive more water than the rest. One of the interviewed farmers at the beginning of a gravity line said that besides tampering with the FTA, he had replaced the pipes receiving water from the FTA with wider pipes—-allowing larger volumes of water to flow to his farm. This, however, deprived farmers downstream of access to their designated water quotas. The resulting inequities in access to water at the farm level served to amplify the immediate water scarcity problem at those farms, which also explains why some of the interviewed farmers did not suffer irrigation water shortages, while counterparts located downstream on the irrigation lines did. This issue became particularly acute after the JVA reduced the water quotas in the NJV. The interviewed farmers reported that their water quotas had been decreased from 48 to 12 h (or less) per week.

Other farmers abstracted water directly from the heavily polluted Jordan River. These farmers also reported use of desalination units to reduce the salt content of the water. This, however, imposed an additional cost, making this option unaffordable for poor farmers, many of whom were solely dependent on the water quota allocation. They faced dire financial consequences due to the worsening water scarcity.

## 4. Discussion

Social and physical intermediaries were identified as playing an important role in enabling or hindering access to and allocation of irrigation water in the NJV (see Figure 1, top right). These intermediaries are interlinked and, furthermore, a result of contestation and conflict between formal and informal arrangements, as described in the results section

above. Table 5 presents the main social and physical intermediaries found in the study area. The sections below then discuss some of these.

**Table 5.** Social and physical intermediaries of irrigation water access and allocation in the Northern Jordan Valley (NJV).

| Social Intermediaries | Physical Intermediaries |
|---|---|
| Perceptions of farmers, donors and the government of water reuse–reallocation plan | New treated wastewater reallocation infrastructure (developed in 2017) |
| Water users associations (WUAs) | Informal practices to access water |
| Jordan Valley Authority (JVA), through canal operators | On-farm water distribution devices ('Farm Turn-out Assembly') |
| Crops farmed and access to national and regional markets | Formal freshwater reallocation infrastructure |
| Experiences of MJV farmers with treated wastewater | On-farm irrigation technology |
| Social networks, influence and financial status of NJV farmers | Farms' geographical location upstream or downstream in irrigation line |
| Government's commitment to secure water supply for urban settings | Physical water scarcity in Jordan |

### 4.1. Perceptions

The experience of MJV farmers with treated wastewater reuse, particularly loss of their citrus plantations, was a key social intermediary that shaped the perceptions of the NJV farmers interviewed in the current research. On the other hand, the views of donor agencies (particularly USAID) and of government regarding the great potential of the water reuse–reallocation plan in the NJV pushed forward the construction of infrastructure and implementation of a few pilots. Understanding perspectives on both sides is instrumental to understand the current stalemate which has halted progress towards putting the reuse–reallocation plan into effect. Table 6 summarizes the drivers of the different stakeholders' perceptions of the reuse–reallocation plan in the NJV.

**Table 6.** Risks and benefits of water reuse as perceived by key stakeholders in the Northern Jordan Valley (NJV).

| | Stakeholder | Perceived Risks | Perceived Benefits/Opportunities |
|---|---|---|---|
| Technical/infrastructural and organization-oriented perceptions | Donor organizations | Impacts on current crops and agricultural practices, potentially undermining farmers' livelihoods | Securing large volumes of freshwater for priority domestic uses in urban settings |
| | Government (e.g., the Ministry of Water, the National Agricultural Research Center and the Jordan Valley Authority) | Considerable resistance exerted by the influential agricultural lobby [33] | Enable government to demonstrate better water resources management to international partners and donors, while helping to secure freshwater for priority urban/domestic uses and increasing the water supply for agriculture through treated wastewater |
| Socio-economic oriented perceptions in the NJV | Small-scale, poor and less influential farmers | Loss of productivity due to limited knowledge and resources to shift from traditional agricultural practices and crop patterns to new techniques and crops, which may be needed due to the shift in water quality | Provision of a reliable irrigation water source, allowing for sufficient water quotas to boost production |
| | Rich farmers | Loss of access to regional and global markets due to health restrictions on crops irrigated with treated wastewater or mixed water stream | None at present |

### 4.2. WUAs and JVA

The roles of the WUAs and JVA have been interlinked, as the WUAs' very establishment is dependent on the JVA. Moreover, JVA regulations govern the WUAs. One of these is the stipulation that for a fully functional WUA, representation of at least 86% of the

agricultural units in the affected area is required. The representation requirement is based on the idea that the local WUA would then represent most landowners in the area. In practice, however, WUA membership rules have favoured conditions in which leadership can be usurped by rich farmers, enabling them to entrench their political influence and potentially consolidate greater access to irrigation water. Hence, WUAs have become yet another form of top-down control over irrigation water, with control of allocations in the hands of the JVA through the JVA's financial and administrative authority over the WUAs. For instance, in case of a breach of water quotas and management stipulations, WUA employees must report the violation to the JVA, as they themselves have no legislative capacity to act. Mustafa et al. (2016) observed that this institutional dependency of the WUAs on the JVA and the current restrictive rules are favoured by many JVA staff, who are concerned they might lose their powerful social and political positions in the Jordan Valley if the WUAs were fully empowered.

### 4.3. Access to Irrigation Water and Access to Markets

Various uncertainties accompany the planned transition from freshwater irrigation to mixed freshwater and treated wastewater for irrigated agriculture in the NJV. The impacts of such a transition could well determine farmers' access to various markets, affecting their local and regional competitiveness, their market share, and the financial and economic viability of the agricultural sector overall. As highlighted by the heterogeneous perceptions of stakeholders (see Table 6), poor and less influential farmers exhibited greater openness to discussing water reuse as a potential source of irrigation water. This was mainly due to their desperate need for additional water to irrigate their lands—as they could not meet this need with the formal freshwater quotas from the KAC or through available informal arrangements, which were either too costly or required social and political influence.

Rich farmers, however, were mainly concerned about losing their access to regional and international export markets, due to the risk of residual contaminants being transferred from treated wastewater to agricultural produce. Fears were raised that the use of treated wastewater for irrigation could trigger a wave of boycotts of Jordan's agricultural products in export markets. Such markets have been known to impose strict regulations on products irrigated with treated wastewater [37], with one example being the ban on agricultural products (particularly tomatoes) from Jordan imposed by Saudi Arabia 21 years ago [38].

The earlier-mentioned USAID report [28] views the "considerable resistance" of local farmers (mainly rich farmers) to the government's water reuse–reallocation plan as a justifiable and predictable response. It also observes that the Jordanian government might be unable to implement such a plan in a top-down way unless further significant reductions in freshwater availability force local farmers to accept water reuse as an inevitable solution to maintain agriculture in the NJV.

Resistance has persisted, however, and translated into various forms of informal irrigation water access. This has resulted in a complex, socially embedded system with practices that have developed independently from those of the formal system [39]. The current study identified two main forms of informal water access for irrigation purposes. The first, used predominantly by poor and less influential farmers, is the adoption of illegal practices to access water for irrigation. These practices were dependent, however, on the availability of alternative water sources (e.g., shallow wells and direct access to the KAC) near a farm's location. The second form of informal water access concerns institutional arrangements exclusive to rich and well-connected farmers and involving multiple actors from governmental institutions in the water sector. Although these informal institutional arrangements were not illegal, they were beyond the reach of poor and less politically and financially influential farmers, who lacked the requisite connections to institutions such as the JVA. Hence, institutional arrangements indeed enabled rich farmers to circumvent the bureaucratic arrangements with impunity [9].

## 5. Conclusions

Many NJV farmers view treated wastewater as a lower quality water resource with many potential negative impacts on agricultural practices and farmers' livelihoods. Similarly, the reallocation of freshwater from the NJV to urban centres (e.g., Amman) is seen as a threat to farmers' identity, power and culture, in favour of "external actors" [40]. Conducting semi-structured interviews and analysing the local dynamics that shape agricultural practices in the NJV and access to irrigation water, the current research found that the involved stakeholders had different and often conflicting perceptions of treated wastewater reuse and its associated risks and benefits (see Table 6). This variation in perspectives mirrored the stakeholders' different socio-political and economic standpoints, leading to the current stalemate blocking the reuse–reallocation plan from being put into operation in the NJV, despite the readiness of the infrastructural component since 2017.

This research supports the hypothesis that the current stalemate is due to the reuse–reallocation plan focusing primarily on the infrastructural and organizational water control aspects, with less attention being given to the local socio-political and economic context in the NJV. As a result, farmers have continued to resist the formal plans, while co-creating informal solutions to access freshwater for irrigation as an adaptive mechanism to overcome immediate water scarcity and maintain agricultural productivity. Although most of the 'informal' practices identified were illegal, the financial gains from increased water availability outweighed the costs incurred, in the form of fines or bribes paid to JVA or WUA staff, particularly among rich farmers who feared losing their access to regional and international markets.

At the same time, poor and less influential farmers—who were in desperate need of water—viewed the treated wastewater reuse plan as an opportunity to address the water scarcity problems they faced on a daily basis, though they too had concerns about the quality of the treated wastewater and its long-term impact on their crops, especially citrus trees. These farmers were open to discussing treated wastewater reuse as a potential source for irrigation water in the NJV. However, their voices have continued to be marginalized, as the current socio-political and economic context empowers rich farmers and their representation through long-established networks and their prominent role in the WUAs.

NJV farmers' adoption of informal practices and institutional arrangements to access water for irrigation has remained beyond the government's capacity to monitor, regulate or prevent. Yet, the potential impact of these practices and arrangements on freshwater consumption is grave and could accelerate water scarcity in the NJV beyond the government's already dire predictions. Continuation of the current stalemate would undermine the government's plan to reallocate freshwater to priority urban and domestic uses, and thus reduce Jordan's options for addressing severe water scarcity in the future. There is an urgent need for the government to develop an inclusive plan addressing farmers' concerns regarding the short- and long-term impacts of wastewater reuse in agriculture. Primarily, policies need to be developed to protect crop exports to regional and international markets and avoid bans on Jordanian agricultural produce such as the one imposed by Saudi Arabia 21 years ago.

In conclusion, this study indicates that understanding the heterogeneous socio-political and economic context—as well as climate change impacts outside the scope of the current paper—is key in formulating water reuse and reallocation policies. Finding a path forward in the NJV will require building a strong alliance with 'ordinary' farmers. The government could furthermore consider restructuring the WUAs to better represent the heterogeneity of NJV farmers' perceptions and needs.

**Author Contributions:** Case study and methodological design: M.H.T. and H.A.-Z.; Theoretical design and conceptualization: M.H.T. and J.H.; Fieldwork and data collection: M.H.T., H.A.-Z. and M.A.-Z.; Data analysis: M.H.T. and H.A.-Z., support J.H.; First draft: M.H.T., H.A.-Z. and J.H., support: J.M.-S., M.A.-Z., A.E. and P.H.; Revisions—first draft: J.H., P.H. and J.M-S.; Revisions—second draft: A.E., J.H., P.H. and J.M.-S.; Maps and tables: M.H.T.; Final revision and editing: M.H.T. and J.H. All authors have read and agreed to the published version of the manuscript.

**Funding:** This research received no external funding.

**Informed Consent Statement:** Informed consent was obtained from all subjects involved in the study.

**Data Availability Statement:** The data of the interviews, field notes and observations are available from the fist author.

**Acknowledgments:** This paper is based on research conducted under the ReWater MENA project, which is led by the International Water Management Institute (IWMI) and funded by the Swedish International Development Cooperation Agency (Sida). Special thanks to Middle East and North Africa regional office (IWMI MENA) and the Royal Scientific Society (RSS) in Amman, Jordan, for facilitating the data collection.

**Conflicts of Interest:** The authors declare no conflict of interest.

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
