# Peer review of "Shifting Waters: The Challenges of Transitioning from Freshwater to Treated Wastewater Irrigation in the Northern Jordan Valley"

_water, doi:10.3390/w15071315_

Round 1

Reviewer 1 Report

Paper is well organized and well written so i recommend this paper for publication however, Minor spelling and sentence correction should be done and cite these articles in revised manuscript.

Comparative Study Between Two Zeolitic Imidazolate Frameworks as Adsorbents for Removal of Organoarsenic, As(III) and As(V) Species from Water,

Design, Synthesis and Spectroscopic Characterizations of Medicinal Hydrazide Derivatives and Metal Complexes of Malonic Ester.

Author Response

I would like to thank the reviewer for the positive feedback. Although, after checking the two recommended articles for citation, we found that the two suggested citations do not match with the scope of our article.

Reviewer 2 Report

Manuscript ID: water-2275649
Title: Shifting waters: the challenges of transitioning from freshwater to treated wastewater irrigation in the Northern Jordan Valley.
OVERVIEW
The paper presents and analyses available data and describes the Northern Jordan Valley. Reports interviews results with farmers and formal stakeholders on the reuse of wastewater for irrigation. Also concludes with a recommendation that highlights the need for an inclusive and participatory approach to bridge the trust gap between farmers and the government and help build an inclusive water reuse plan with minimized risks to farmers.  
GENERAL COMMENTS
The subject matter is actual, interesting and within the scope of the Journal Water.
The manuscript complies with the journal template.
The title is adequate.
The English spelling and grammar are fine.
The manuscript is original, and plagiarism was not detected.
The objectives are clearly stated.
The manuscript provides information on its replicability and reproducibility.
The analyses are appropriate and well-described.
The tables and figures are fine.
The interpretation and results are supported by the data.
The conclusions report the major findings of the study.
The strengths and limitations of the study are reported.
The manuscript structure, flow and writing are fine.
The manuscript addresses a well-known problem and reports the local status quo of irrigation water use in the Northern Jordan Valley.
In conclusion, I believe this manuscript is interesting and worthy of publication after minor changes. Please read the specific comments.
SPECIFIC COMMENTS
Across the manuscript, the farmers are grouped into poor farmers and rich farmers. Please present the quantitative criterium used by the authors to define a poor farmer and a rich farmer.  
Line 98: Please explain the meaning of the “adjacent to irrigation lines 11, 12 and 13”? If possible, include a map.
Lines 144, 151, 171, 221, 300, 356, 361, 363, 372, 432, 438, 449, 476, 518: Please correct the broken links “Error! Reference source not found”.
Line 367, Figure 3: Please add a legend explaining the current and the proposed water reallocation fluxes.

Author Response

I would like to thank the reviewer for the constructive review.

  • Regarding the criterium used to differentiate between "rich" and "poor" farmers - this is mainly two things: the area of land owned by the farmer (i.e. number of farm units) and the agricultural / irrigation techniques used by the farmer. where more privileged (or rich) farmers in the Northern Jordan Valley tend to shift to drip irrigation and install solar panels, this is in addition to their access to the export market (this was identified from the literature).
  • Irrigation lines in the Northern Jordan Valley are numbered chronologically starting from the Northmost irrigation pump station. For this article, we have interviewed farmers who own/manage farm units that are irrigated via irrigation lines 11, 12 and 13. 
  • Broken links “Error! Reference source not found” are now fixed.
  • a legend has been added to explain the current and proposed reallocation schemes

Reviewer 3 Report

This study outlines a plan to address water scarcity in Jordan through treated wastewater reuse, which has faced farmer resistance. The research recommends an inclusive approach to bridge the trust gap between farmers and the government. In general, this manuscript is meaningful and ready to publish, with the exception of a few errors as follows:

Line 120: could you give an example what is Informal arrangement?

Line 144, 151: no reference source found

Line 171: no reference source found

Line 221: no reference source found

Line 300-301, 356-363: no reference source found

Line 432, 438, 476: no reference source found

Author Response

I would like to thank the reviewer for the constructive feedback.

regarding the first comment: There are examples of the informal arrangements under the sub-title "Informal access to water for irrigation" line 395.

Regarding the rest of the comments: All the broken cross-references have been fixed in the revised version.

Reviewer 4 Report

The article “Shifting waters: the challenges of transitioning from freshwater to treated wastewater irrigation in the Northern Jordan Valley” analyzes the socio-economic causes of farmer resistance and contestation as concerns treated wastewater reuse. The general topic of the article is in the interest of Water journal; however, despite being novel, the proposed content appears too qualitative, and several integrations are needed to reach a high-quality standard for publication. The English language is poor. I subsequently recommend a major revision that should cover the following points:

1.      Introduction: the concept of fertigation (combined water and nutrient recovery) should be introduced as well in the section. Moreover, recently remarkable review papers have been published concerning wastewater reuse in agriculture; they should be considered to broaden the overview and better set the general framework (see 10.1016/j.envpol.2021.118755 and 10.3390/su9101734).

2.      The titles of sections and subsections should have progressive numbering (see MDPI standards).

3.      Some references are not correctly linked throughout the whole manuscript (e.g., lines 144 and 151).

4.      Materials and methods: this section should quantitatively report how the questionnaires were conducted, how much data were collected, and how the results were elaborated in order to make the conducted study reproducible. The section currently appears too qualitative, I suggest revising it thoroughly deleting the parts that are not useful for reproducing the study, and focusing instead on the most important and concrete aspects to be delivered to the reader.

5.      Material and methods, sub-section “Study area”. It would be interesting to report some basic physicochemical characteristics of the treated effluents from the As-Samra plant, comparing them to the required legislation standards. This would make the section more quantitative and interesting to the readers.

6.      Lines 213-218 appear to be pertinent to Materials and Methods, rather than to the Results section.

7.      Table 2 is a bit chaotic, I would reorganize it in a clearer way.

8.      Results: again, I would prefer a more quantitative section, with Tables and Graphs having a higher value for the reader. The section is very qualitative, and some content appears repetitive. A thorough revision is needed.

9.      Results, sub-section “Water control infrastructure”: again, it is important to mention some mean physicochemical characteristics of the treated effluents the from As-Samra plant to understand if the effluent may effectively alter the quality of the irrigated crops.

10.   Conclusions sections should be shortened at least by 25-30%, focusing on the key points to be delivered to the reader.

11.   The English language is poor and needs significant improvement to reach a high-quality standard for publication.

Round 2

Reviewer 4 Report

The authors addressed in a good way reviewer comments. The manuscript can thus be accepted for publication.